# A Sensor Fusion Based Nonholonomic Wheeled Mobile Robot for Tracking Control

**DOI:** 10.3390/s20247055

**Published:** 2020-12-09

**Authors:** Shun-Hung Tsai, Li-Hsiang Kao, Hung-Yi Lin, Ta-Chun Lin, Yu-Lin Song, Luh-Maan Chang

**Affiliations:** 1Graduate Institute Automation Technology, National Taipei University of Technology, Taipei 10608, Taiwan; shtsai@ntut.edu.tw (S.-H.T.); t109669011@ntut.edu.tw (L.-H.K.); eugenesong1130@gmail.com (T.-C.L.); 2High-Tech Facility Research Center, Department of Civil Engineering, National Taiwan University, Zhubei 30264, Taiwan; hungyilin@ntu.edu.tw (H.-Y.L.); luhchang@ntu.edu.tw (L.-M.C.); 3Department of Bioinformatics and Medical Engineering, Asia University, Taichung 41354, Taiwan

**Keywords:** nonholonomic, wheeled mobile robot (WMR), tracking control, global positioning system (GPS), radio frequency (RF), Kalman filter, quaternion, trajectory tracking

## Abstract

In this paper, a detail design procedure of the real-time trajectory tracking for the nonholonomic wheeled mobile robot (NWMR) is proposed. A 9-axis micro electro-mechanical systems (MEMS) inertial measurement unit (IMU) sensor is used to measure the posture of the NWMR, the position information of NWMR and the hand-held device are acquired by global positioning system (GPS) and then transmit via radio frequency (RF) module. In addition, in order to avoid the gimbal lock produced by the posture computation from Euler angles, the quaternion is utilized to compute the posture of the NWMR. Furthermore, the Kalman filter is used to filter out the readout noise of the GPS and calculate the position of NWMR and then track the object. The simulation results show the posture error between the NWMR and the hand-held device can converge to zero after 3.928 seconds for the dynamic tracking. Lastly, the experimental results show the validation and feasibility of the proposed results.

## 1. Introduction

Wheeled mobile robot (WMR) has been utilized in many fields such as the automated guided vehicle (AGV) [1,2], robotic cleaner [3], exploration robot [4,5] recently. The WMR has the capability to respond flexibility by itself in various complex unknown environments. 9-axis IMU sensor integrated with a three-axis accelerometer, a three-axis gyroscope, and a three-axis magnetometer is used to estimate linear velocity, angular velocity and orientation of the body relative to the global reference frame. Hence, the posture of the body can be obtained [6,7,8]. The free, open, and dependable nature of GPS has led to the development of applications affecting every aspect of modern life. GPS is capable of receiving signal from GPS satellites and someone can use these useful received information to calculate the geographical position of the body [9,10].

Different types of the WMR had been proposed. Omnidirectional WMR (OWMR) is one of the WMR that has the advantage of multimode action and high mobility [11,12,13,14,15]. NWMR is another type of the WMR. Owing to the nonholonomic constraint, the moving characteristic of the NWMR is rolling but not sliding, as a result, NWMR can not move laterally and difficult to turn the body in any direction. The goal of the trajectory tracking is to design a motion controller of WMR for adjusting linear velocity and angular velocity, so the WMR can track to the desired trajectory. The aim of the tracking controller is to minimize the tracking errors between the real trajectory and the desired trajectory. These errors are arisen from slippage, disturbances and the measured error of sensor readout. A cross-coupled controller is presented in [16]. The controller is used to reduce wheel slip of the vehicle. Sun and Yan presented an evaluation method to estimate the unknown disturbance. The estimated information is compensated by the controller of the NWMR, hence, the tracking error of the NWMR is repressed [17,18]. In [19], Rayguru proposed a robust-observer based sliding mode controller to achieve the motion control task in the presence of incomplete state measurements and sensor inaccuracies. A two-layer lateral path tracking controller are presented in [20], the upper-layer controller is implemented with a linear time-varying model predictive control (LTV-MPC) algorithm, the lower-layer controller is implemented by a radial basis function neural network proportion-integral-derivative (RBFNN-PID) algorithm. The advantage of the two-layer controller is that controller can track the reference paths accurately while ensuring the stability of the vehicle, however, this type of the tracking controller suffers from high computation. In [21], a single controller for simultaneous stabilization and trajectory tracking of NWMR is proposed. A controller with time-varying parameters is designed to stabilize the error system. Then, the geometric analysis method is used to guarantee that the controller inputs stay in the restricted input domain. Hence the controller can prevent the actuator from saturation and the stability of the system is improved. Kanayama and Sanhoury presented the design of the tracking controllers with stability [22,23]. These two types of tracking controller based on the direct Lyapunov method are used for solving the lateral error issue. The advantages of these two tracking controllers are less computational complexity and fast response to the tracking error.

The contribution of this paper is to implement the real-time trajectory tracking for NWMR. IMU, GPS and RF sensors are integrated into NWMR. In order to be implemented into the resource limited embedded system, a low computational complexity, fast response to tracking error and stability of tracking controller were key factors and need be traded off. Hence, the conventional Kalman filter [24] and tracking controller proposed by [23] are utilized. The rest of this paper is organized as follows: Section 2 illustrates the system function block and system flow chart of proposed NWMR. Section 3 presents the kinematic model of the NWMR, the design of the tracking controller on the NWMR is discussed. In Section 4, the algorithms for the sensors equipped on the NWMR are introduced. Section 5 demonstrates the simulations and experimental results of the developed NWMR. Finally, conclusions and future works are given in Section 6.

## 2. The Proposed Architecture

In this section, the proposed NWMR is described in Section 2.1. Section 2.2 illustrates the system flow chart of the NWMR.

### 2.1. The Block Diagram

The block diagram of the proposed NWMR system is shown in Figure 1. In the NWMR, a 9-axis sensor is utilized to acquire the current posture of the robot. Two GPS modules are used to obtain the current positions of the NWMR and the target hand-held device, respectively. RF transmitter in the hand-held device transmits the position information of the tracking object to the NWMR. The tracking controller and sensor fusion algorithms are implemented in the STM32F429 microprocessor [25].

### 2.2. The Flow Charts of Trajectory Tracking

Figure 2 shows the system flow chart of the NWNR. The current position data of NWNR is obtained by GPS and is processed by the Kalman filter. The current posture data of NWNR is acquired by the 9-axis IMU sensor and then is processed by the Madgwick’s data fusion algorithm [26,27]. RF module receive the current position data from target hand-held device. Using these three data as input parameters of the tracking controller, the tracking controller calculates the shortest distance from NWNR to the hand-held device. While the distance between the NWNR and the hand-held device is smaller than 1.5 m, the tracking controller will stop the tracking task, and the mission of trajectory tracking is finished.

Figure 3 shows the system flow chart of the hand-held device which is considered as the tracking object. The position data of hand-held device is obtained by GPS and is refined by the the Kalman filter. After processing, the data is sent out by the RF module.

## 3. Kinematic Model and Tracking Controller Design

As depicted in Figure 4, the NWMR considered in this paper is a kind of four-wheeled robot. This robot body has symmetric shape and the center of mass is at the geometric center C of the body. Two rear wheels are driven differentially by motors, two front wheels prevent the NWMR from tipping over while the robot moves on a plane. In this paper, it is assumed that the motion of two front wheels can be ignored in dynamics of the NWMR.

### 3.1. Kinematic Model of the NWMR and Tracking Problem Representation

Figure 4 shows the posture of the NWMR in the global X-Y coordinate. The local coordinate system is fixed to the NWMR with point C as the origin. ($x_{c}$,$y_{c}$) is the current position of the geometric center C in the global X-Y coordinate, $\theta_{c}$ is the angle between the X-axis and $X_{r}$ -axis, it is represented as the heading direction of the NWMR, $v_{c}$ denotes the linear velocity of the NWMR in the direction of $X_{r}$ -axis and $\omega_{c}$ is the angular velocity of the robot.

Consider the kinematic model of the NWMR, the posture variable of NWMR, $r_{c}$, is defined as
(1)$$r_{c}={\left[x_{c}\;\;y_{c}\;\;\theta_{c}\right]}^{T}$$

The moving of the robot is controlled by $v_{c}$ and $\omega_{c}$, the input state variable of the robot, $u_{c}$, is therefore defined as
(2)$$u_{c}={\left[v_{c}\;\;\omega_{c}\right]}^{T}$$

Under the nonholonomic constraint,
(3)$${\dot{x}}_{c}\sin\theta_{c}-{\dot{y}}_{c}\cos\theta_{c}=0$$

The kinematic model of the NWMR can be expressed by [28]
(4)$${\dot{r}}_{c}=\left[\begin{array}{c} {\dot{x}}_{c}\\ {\dot{y}}_{c}\\ {\dot{\theta }}_{c} \end{array}\right]=\left[\begin{array}{cc} \cos\theta_{c} & 0\\ \sin\theta_{c} & 0\\ 0 & 1 \end{array}\right]u_{c}=\left[\begin{array}{c} v_{c}\cos\theta_{c}\\ v_{c}\sin\theta_{c}\\ \omega_{c} \end{array}\right]$$

The position information of the tracking object is obtained by the hand-held device via GPS module. As shown in Figure 5, it is assumed that the hand-held device is circular shape and the center of mass is located at the geometric center D of the device.

($x_{d}$,$y_{d}$) is the current position of the geometric center D in the X-Y coordinate. $\theta_{d}$ is the angle between the X-axis and the straight line that pass through point C and point D, which is defined as
(5)$$\theta_{d}=\frac{180}{\pi }tan^{-1}\left(\frac{\left|y_{d}-y_{c}\right|}{\left|x_{d}-x_{c}\right|}\right)$$

The posture of the hand-held device, $r_{d}$, is defined as
(6)$$r_{d}={\left[x_{d}\;\;y_{d}\;\;\theta_{d}\right]}^{T}$$

For simplicity, it is assumed that the nonholonomic constraint of the hand-held device can be written as
(7)$${\dot{x}}_{d}\sin\theta_{d}-{\dot{y}}_{d}\cos\theta_{d}=0$$

Two postures are used in this tracking control system; One is current posture, $P_{c}$, and the other is reference posture, $P_{d}$. Current posture is the real posture of the NWMR, reference posture is the target posture, i.e., the real posture of the hand-held device, hence
(8)$$P_{c}=r_{c}\;\;;\;\;P_{d}=r_{d}$$

As illustrated in Figure 6, the posture error, $P_{e}$, between the reference posture and the current posture in the local $X_{r}-Y_{r}$ coordinate is expressed as
(9)$$P_{e}=\left[\begin{array}{c} x_{e}\\ y_{e}\\ \theta_{e} \end{array}\right]=\left[\begin{array}{ccc} \cos\theta_{c} & \sin\theta_{c} & 0\\ -\sin\theta_{c} & \cos\theta_{c} & 0\\ 0 & 0 & 1 \end{array}\right]\left(P_{d}-P_{c}\right)=R\left(\theta \right)\left(P_{d}-P_{c}\right)$$
where $x_{e}$ is the error in the $X_{r}$ direction of the robot, $y_{e}$ is the error in the $Y_{r}$ direction of the robot, $\theta_{e}$ is the orientation error, i.e., $\theta_{e}=\theta_{d}-\theta_{c}$, R(θ) is rotation matrix and it is expressed as the orientation of the current posture, $P_{c}$, with respect to the reference posture, $P_{d}$.

By differentiating (Equation 9), we have
(10)$$\begin{array}{cc} {\dot{x}}_{e} & =\left({\dot{x}}_{d}-{\dot{x}}_{c}\right)\cos\theta_{c}+\left({\dot{y}}_{d}-{\dot{y}}_{c}\right)\sin\theta_{c}-\left(x_{d}-x_{c}\right){\dot{\theta }}_{c}\sin\theta_{c}+\left(y_{d}-y_{c}\right){\dot{\theta }}_{c}\cos\theta_{c}\\ \; & ={\dot{x}}_{d}\cos\theta_{c}+{\dot{y}}_{d}\sin\theta_{c}-\left(v_{c}\cos^{2}\theta_{c}+v_{c}\sin^{2}\theta_{c}\right)+y_{e}\omega_{c}\\ \; & ={\dot{x}}_{d}\cos\theta_{c}+{\dot{y}}_{d}\sin\theta_{c}-v_{c}+y_{e}\omega_{c}\\ \; & ={\dot{x}}_{d}\cos\left(\theta_{d}-\theta_{e}\right)+{\dot{y}}_{d}\sin\left(\theta_{d}-\theta_{e}\right)-v_{c}+y_{e}\omega_{c} \end{array}$$

Substituting (Equation 7) and the kinematic model of the NWMR at point D into (Equation 10), we obtain
(11)$${\dot{x}}_{e}=v_{d}\cos\theta_{e}-v_{c}+y_{e}\omega_{c}$$

Similarly,
(12)$$\begin{array}{cc} {\dot{y}}_{e} & =-\left({\dot{x}}_{d}-{\dot{x}}_{c}\right)\sin\theta_{c}+\left({\dot{y}}_{d}-{\dot{y}}_{c}\right)\cos\theta_{c}-\left(x_{d}-x_{c}\right){\dot{\theta }}_{c}\cos\theta_{c}-\left(y_{d}-y_{c}\right){\dot{\theta }}_{c}\sin\theta_{c}\\ \; & =-{\dot{x}}_{d}\sin\theta_{c}+{\dot{y}}_{d}\cos\theta_{c}-x_{e}\omega_{c}\\ \; & =v_{d}\sin\theta_{e}-x_{e}\omega_{c} \end{array}$$
(13)$${\dot{\theta }}_{e}={\dot{\theta }}_{d}-{\dot{\theta }}_{c}=\omega_{d}-\omega_{c}$$

Combining (11) to (13), the differential equation of the posture error of NWMR with respect to the hand-held device is obtained
(14)$${\dot{P}}_{e}=\left[\begin{array}{c} {\dot{x}}_{e}\\ {\dot{y}}_{e}\\ {\dot{\theta }}_{e} \end{array}\right]=\left[\begin{array}{c} v_{d}\cos\theta_{e}-v_{c}+y_{e}\omega_{c}\\ v_{d}\sin\theta_{e}-x_{e}\omega_{c}\\ \omega_{d}-\omega_{c} \end{array}\right]$$

### 3.2. The Tracking Controller Design

The function block of the tracking controller of the NWMR is demonstrated in Figure 7. Firstly, the posture error $P_{e}\left(t\right)$ between $P_{d}\left(t\right)$ and $P_{c}\left(t\right)$ at time *t* is obtained by rotation matrix from (Equation 9). Based on the variable $P_{e}\left(t\right)$, the tracking controller is designed to stabilize the kinematic model of NWMR in concurrent with unknown error or disturbances $N_{noise}$. The variable $u_{d}\left(t\right)={\left[v_{d}\left(t\right)\;\;\omega_{d}\left(t\right)\right]}^{T}$ are the linear velocity and the angular velocity of the reference posture and is acted as the input of the tracking controller. Finally, the updated posture $P_{c}\left(t\right)$ of the NWMR generated by an integrator is used for next posture error correction.

The tracking controller proposed by [23] is used for the developed NWMR, we have
(15)$$u_{c}=\left[\begin{array}{c} v_{c}\\ \omega_{c} \end{array}\right]=\left[\begin{array}{c} v_{d}\cos\theta_{e}+k_{1}x_{e}+k_{4}sign\left(x_{e}\right)y_{e}^{2}\\ \omega_{d}+v_{d}\left(k_{2}y_{e}+k_{3}\sin\theta_{e}\right) \end{array}\right]$$
where $k_{1}$, $k_{2}$, $k_{3}$ and $k_{4}$ are adjustable gain coefficients, $sign(x_{e})$ is described as:(16)$$sign\left(x_{e}\right)=\left\{\begin{array}{c} -1,\;x_{e}<0\\ \;\;\;1,\;x_{e}\geq 0 \end{array}\right.$$

By using Lyapunov function and Routh-Hurwitz Criterion to examine the stability of the tracking controller, the posture error $P_{e}=0$ is uniformly asymptotical stable over interval [0,∞) under the conditions: (a) $v_{d}$ and $\omega_{d}$ are continuous, (b) $v_{d}$, $\omega_{d}$, $k_{1}$, $k_{2}$, $k_{3}$ and $k_{4}$ are all bounded and (c) ${\dot{v}}_{d}$ and ${\dot{\omega }}_{d}$ are sufficiently small [22].

## 4. Signal Processing Algorithms

In this section, the sensor fusion algorithms for the NWMR and hand-held device are introduced.

### 4.1. Madgwick’s Data Fusion Algorithm for the 9-Axis IMU Sensor

The accurate measurement of orientation plays a key factor in the design of the tracking control of the NWMR. A MEMS gyroscope outputs angular velocity and the data is used to compute the orientation of the robot. However, this sensor suffers from an accumulated error. An accelerometer and magnetometer measure the earth’s gravity and magnetic field, respectively. These two sensors provide an absolute reference of orientation. Nevertheless, these sensors subject to high levels of noise. The goal of the data fusion algorithm is to estimate a single orientation through the optimal data fusion of gyroscope, accelerometer and magnetometer measurements.

Euler angles are three angles and can be used to describe the orientation of a mobile frame of reference. However, in some motion case, Euler angles suffer from gimbal lock mechanism [29]. Quaternions developed by Hamilton is a mathematical number system that can be treated as the other representation of the three-dimensional space [30]. Quaternion method is well used in mechanics, computer graphics, crystallographic texture analysis in three-dimensional space for preventing from the gimbal lock mechanism of the Euler angles. In inertial navigation system (IMS), it uses quaternion methods to derive the orientation and the velocity of vehicle. In this paper, we also use quaternion method to calculate three-dimensional motion of the NWMR.

The data fusion algorithm proposed by Madgwick is applied for the 9-axis [26,27]. Figure 8 shows the block diagram of the Madgwick’s fusion algorithm. The measured output data of the 3-axis sensor ${}_{}^{S}\hat{d}$ in the sensor frame can be transfer to the earth frame ${}_{}^{E}\hat{d}$ by using a quaternion ${}_{E}^{S}\hat{q}$, yields
(17)$$\begin{array}{cc} \; & {}_{}^{E}\hat{d}={}_{E}^{S}\hat{q}⊗{}_{}^{S}\hat{d}⊗{}_{E}^{S}{\hat{q}}^{*}\\ \; & {}_{}^{E}\hat{d}=\left[0\;d_{x}\;d_{y}\;d_{z}\right]\\ \; & {}_{E}^{S}\hat{q}=\left[q_{1}\;\;q_{2}\;q_{3}\;\;q_{4}\right]\\ \; & {}_{S}^{E}{\hat{q}}^{*}={}_{E}^{S}\hat{q}=\left[q_{1}\;\left(-q_{2}\right)\;\left(-q_{3}\right)\;\left(-q_{4}\right)\right] \end{array}$$
where $d_{x}$, $d_{y}$ and $d_{z}$ are the three dimensional vector components of ${}_{}^{E}\hat{d}$, which aligns a predefined reference direction in the earth frame, ⊗ is quaternion product, $q_{1}\;\;q_{2}\;\;q_{3}$ and $q_{4}$ are the components of ${}_{E}^{S}\hat{q}$, and ${}_{E}^{S}{\hat{q}}^{*}$ is conjugate of ${}_{E}^{S}\hat{q}$. A 3-axis gyroscope measures the angular velocity of robot, the angular rate ${}_{}^{S}\omega$ is define as
(18)$${}_{}^{S}\omega =\left[0\;\;{}_{}^{S}\omega_{x}\;\;{}_{}^{S}\omega_{y}\;\;{}_{}^{S}\omega_{z}\right]$$
where ${}_{}^{S}\omega_{x}$, ${}_{}^{S}\omega_{y}$ and ${}_{}^{S}\omega_{z}$ is the angular velocity of the X, Y, Z axes in the sensor frame, respectively. Derivative of the quaternion at time *t* describes the rate of the change of orientation,${}_{E}^{S}{\dot{\hat{q}}}_{\omega \_est}\left(t\right)$, and can be calculated as
(19)$${}_{E}^{S}{\dot{\hat{q}}}_{\omega \_est}\left(t\right)=\frac{1}{2}{}_{E}^{S}{\hat{q}}_{est}\left(t-1\right)⊗{}_{}^{S}\omega \left(t\right)$$
where ${}_{E}^{S}{\hat{q}}_{est}\left(t-1\right)$ is prior estimate of orientation, the sub-script ω indicates that the quaternion is calculated from gyroscope and ${}_{}^{S}\omega \left(t\right)$ is angular velocity measured at time *t*. If the angular velocity of the sensor is not constant, the derivative of quaternion in (Equation 19) can be used to estimate the current orientation of the robot, which is shown as
(20)$${}_{E}^{S}{\hat{q}}_{\omega \_est}\left(t\right)={}_{E}^{S}{\hat{q}}_{est}\left(t-1\right)+{}_{E}^{S}{\dot{\hat{q}}}_{\omega \_est}\left(t\right)\Delta t$$
where Δt is sampling period. Substituting (Equation 19) into (Equation 20), yields
(21)$${}_{E}^{S}{\hat{q}}_{\omega \_est}\left(t\right)={}_{E}^{S}{\hat{q}}_{est}\left(t-1\right)+\frac{1}{2}{}_{E}^{S}{\hat{q}}_{est}\left(t-1\right)⊗{}_{}^{S}\omega \left(t\right)\Delta t$$

Contrary to the orientation calculated from the angular velocity, gravity and magnetic field can also be applied to estimate orientation. Using (Equation 17) and quaternion operations, the measured direction vector in the earth frame can also be rotated to the sensor frame, therefore
(22)$${}_{}^{S}\hat{d}={}_{E}^{S}{\hat{q}}^{*}⊗{}_{}^{E}\hat{d}⊗{}_{E}^{S}\hat{q}$$

In order to estimate the quaternion, the difference between a known vector in the sensor frame and a measured vector in sensor frame is minimized. In other words,
(23)$$min\;f\left({}_{}^{S}{\hat{d}}_{est},{}_{}^{S}\hat{S}\right)=min\;f\left({}_{E}^{S}{\hat{q}}_{est}^{*}⊗{}_{}^{E}\hat{d}⊗{}_{E}^{S}{\hat{q}}_{est}-{}_{}^{S}\hat{S}\right)=min\;f\left({}_{E}^{S}{\hat{q}}_{est},{}_{}^{E}\hat{d},{}_{}^{S}\hat{S}\right)$$
where ${}_{}^{S}{\hat{d}}_{est}$ is a sensor frame’s estimated vector, ${}_{E}^{S}{\hat{q}}_{est}$ is the estimated quaternion and ${}_{}^{S}\hat{S}$ is measured vector in sensor frame. Madgwick uses the gradient descent algorithm to iterate the estimated quaternion ${}_{E}^{S}{\hat{q}}_{est}\left(t\right)$ based on an initial guess quaternion ${}_{E}^{S}{\hat{q}}_{est}\left(0\right)$ and a step-size μ [26,27], the ${}_{E}^{S}{\hat{q}}_{est}\left(t\right)$ can be expressed as
(24)$${}_{E}^{S}{\hat{q}}_{est}\left(t\right)={}_{E}^{S}{\hat{q}}_{est}\left(t-1\right)-\mu \frac{\nabla f({}_{}^{S}{\hat{d}}_{est}\left(t\right),{}_{}^{S}\hat{S}\left(t\right))}{∥\nabla f\left({}_{}^{S}{\hat{d}}_{est}\left(t\right),{}_{}^{S}\hat{S}\left(t\right)\right)∥}$$
(25)$$\nabla f\left({}_{}^{S}{\hat{d}}_{est}\left(t\right),{}_{}^{S}\hat{S}\left(t\right)\right)=J^{T}\left({}_{E}^{S}{\hat{q}}_{est}\left(t-1\right),{}_{}^{E}\hat{d}\left(t\right)\right)f\left({}_{E}^{S}{\hat{q}}_{est}\left(t-1\right){}_{}^{E}\hat{d}\left(t\right),{}_{}^{S}\hat{S}\left(t\right)\right)$$
where (Equation 25) computes the gradient of the solution surface defined by objective function f(·) and its Jacobian J(·).

A 3-axis accelerometer measures not only the magnitude and direction of the gravity in the sensor frame but also linear velocity due to motion of the sensor. A 3-axis magnetometer measures the magnitude and direction of the earth’s magnetic field in the sensor frame compounded with distortion. Following the derivation from (Equation 2) to (Equation 25), the estimated quaternion ${}_{E}^{S}{\hat{q}}_{acc,mag\_est}\left(t\right)$ at time *t* can be calculate by combining the output data of accelerometer and the magnetometer, we have
(26)$${}_{E}^{S}{\hat{q}}_{acc,mag\_est}\left(t\right)={}_{E}^{S}{\hat{q}}_{est}\left(t-1\right)-\kappa \frac{\nabla f_{acc,mag}}{∥\nabla f_{acc,mag}∥}$$
(27)$$\nabla f_{acc,mag}=J_{acc,mag}^{T}\left({}_{E}^{S}{\hat{q}}_{est}\left(t-1\right),{}_{}^{E}{\hat{d}}_{acc,mag}\left(t\right)\right)f\left({}_{}^{S}{\hat{d}}_{acc,mag\_est}\left(t\right),{}_{}^{S}{\hat{S}}_{acc,mag}\left(t\right)\right)$$
(28)$$J_{acc,mag}^{T}\left({}_{E}^{S}{\hat{q}}_{est}\left(t-1\right),{}_{}^{E}{\hat{d}}_{acc,mag}\left(t\right)\right)=\left[\begin{array}{c} J_{acc}^{T}\left({}_{E}^{S}{\hat{q}}_{est}\left(t-1\right),{}_{}^{E}{\hat{d}}_{acc}\left(t\right)\right)\\ J_{mag}^{T}\left({}_{E}^{S}{\hat{q}}_{est}\left(t-1\right),{}_{}^{E}{\hat{d}}_{mag}\left(t\right)\right) \end{array}\right]$$
(29)$$\begin{array}{cc} f({}_{}^{S}{\hat{d}}_{acc,mag\_est}\left(t\right),{}_{}^{S}{\hat{S}}_{acc,mag}\left(t\right))\; & =\left[\begin{array}{c} f({}_{}^{S}{\hat{d}}_{acc\_est}\left(t\right),{}_{}^{S}{\hat{S}}_{acc}\left(t\right))\\ f({}_{}^{S}{\hat{d}}_{mag\_est}\left(t\right),{}_{}^{S}{\hat{S}}_{mag}\left(t\right)) \end{array}\right]\\ \; & =\left[\begin{array}{c} f({}_{E}^{S}{\hat{q}}_{est}\left(t-1\right),{}_{}^{E}{\hat{d}}_{acc}\left(t\right),{}_{}^{S}{\hat{S}}_{acc}\left(t\right))\\ f({}_{E}^{S}{\hat{q}}_{est}\left(t-1\right),{}_{}^{E}{\hat{d}}_{mag}\left(t\right),{}_{}^{S}{\hat{S}}_{mag}\left(t\right)) \end{array}\right] \end{array}$$

The estimated quaternion ${}_{E}^{S}{\hat{q}}_{acc,mag\_est}\left(t\right)$ is achieved based on a previous estimate of quaternion ${}_{E}^{S}{\hat{q}}_{est}\left(t-1\right)$ and the objective function $\nabla f_{acc,mag}$. This objective function is defined by measured data of accelerometer ${}_{}^{S}{\hat{S}}_{acc}\left(t\right)$ and magnetometer ${}_{}^{S}{\hat{S}}_{mag}\left(t\right)$ in sensor frame.

Using the concept of the complementary filter presented in [31], the final estimated quaternion ${}_{E}^{S}{\hat{q}}_{est}\left(t\right)$ is obtained through the fusion of the two estimated quaternions ${}_{E}^{S}{\hat{q}}_{\omega \_est}\left(t\right)$ and ${}_{E}^{S}{\hat{q}}_{acc,mag\_est}\left(t\right)$,
(30)$${}_{E}^{S}{\hat{q}}_{est}\left(t\right)=\left(1-\gamma \right){}_{E}^{S}{\hat{q}}_{\omega \_est}\left(t\right)+\left(\gamma \right){}_{E}^{S}{\hat{q}}_{acc,mag\_est}\left(t\right)$$
where (1−γ) and γ are weights of ${}_{E}^{S}{\hat{q}}_{\omega \_est}\left(t\right)$ and ${}_{E}^{S}{\hat{q}}_{acc,mag\_est}\left(t\right)$, respectively. The convergence rate of ${}_{E}^{S}{\hat{q}}_{acc,mag\_est}\left(t\right)$ is defined as $\frac{\kappa }{\Delta t}$ and the divergence rate of ${}_{E}^{S}{\hat{q}}_{\omega \_est}\left(t\right)$ is defined as β. Following the derivation in [26], the optimal value of γ is defined as
(31)$$\gamma =\frac{\beta }{\frac{\kappa }{\Delta t}+\beta }$$

It is assumed that κ is large, (Equation 26) and (Equation 31) can be rewritten
(32)$${}_{E}^{S}{\hat{q}}_{acc,mag\_est}\left(t\right)\approx -\kappa \frac{\nabla f_{acc,mag}}{∥\nabla f_{acc,mag}∥}$$
(33)$$\gamma =\frac{\beta \Delta t}{\kappa }\approx 0$$

Substituting (Equation 20), (Equation 21), (Equation 32) and (Equation 33) into (Equation 30), we have
(34)$$\begin{array}{cc} {}_{E}^{S}{\hat{q}}_{est}\left(t\right) & =\left(1-\gamma \right){}_{E}^{S}{\hat{q}}_{\omega \_est}\left(t\right)+\left(\gamma \right){}_{E}^{S}{\hat{q}}_{acc,mag\_est}\left(t\right)\\ \; & =\left(1-0\right){}_{E}^{S}{\hat{q}}_{\omega \_est}\left(t\right)+\frac{\beta \Delta t}{\kappa }\left(-\kappa \frac{\nabla f_{acc,mag}}{∥\nabla f_{acc,mag}∥}\right)\\ \; & ={}_{E}^{S}{\hat{q}}_{est}\left(t-1\right)+{}_{E}^{S}{\dot{\hat{q}}}_{\omega \_est}\left(t\right)\Delta t-\beta \frac{\nabla f_{acc,mag}}{∥\nabla f_{acc,mag}∥}\Delta t\\ \; & ={}_{E}^{S}{\hat{q}}_{est}\left(t-1\right)+\frac{1}{2}{}_{E}^{S}{\hat{q}}_{est}\left(t-1\right)⊗{}_{}^{S}\omega \left(t\right)\Delta t-\beta \frac{\nabla f_{acc,mag}}{∥\nabla f_{acc,mag}∥}\Delta t \end{array}$$

From (Equation 34), it can be seen the fusion algorithm minimizes β for the measured magnetic field will be distorted by the presence of ferromagnetic elements in the vicinity of the magnetometer. Due to the error, the estimated direction vector of the earth’s magnetic field ${}_{}^{E}{\hat{d}}_{mag\_est}\left(t\right)$ has nonzero Y-axis component, the ${}_{}^{E}{\hat{d}}_{mag\_est}\left(t\right)$ is calculated as
(35)$$\begin{array}{cc} {}_{}^{E}{\hat{d}}_{mag\_est}\left(t\right) & =[0\;\;{}_{}^{E}d_{mag\_x\_est}\left(t\right)\;\;{}_{}^{E}d_{mag\_y\_est}\left(t\right)\;\;{}_{}^{E}d_{mag\_z\_est}\left(t\right)]\\ \; & ={}_{E}^{S}{\hat{q}}_{est}\left(t-1\right)⊗{}_{}^{S}{\hat{d}}_{mag}\left(t\right)⊗{}_{E}^{S}{\hat{q}}_{est}^{*}\left(t-1\right) \end{array}$$
where ${}_{}^{E}d_{mag\_y\_est}$ is nonzero Y-axis component in the earth frame. These nonzero Y-axis component can be corrected by normalizing to have only X and Z-axes components of the earth frame [26]. Hence
(36)$${}_{}^{E}{\hat{d}}_{mag\_comp\_est}\left(t\right)=\left[0\;\;\sqrt{{}_{}^{E}d_{mag\_x\_est}^{2}\left(t\right)+{}_{}^{E}d_{mag\_y\_est}^{2}\left(t\right)}\;\;0\;\;{}_{}^{E}d_{mag\_z\_est}\left(t\right)\right]$$
where ${}_{}^{E}{\hat{d}}_{mag\_comp\_est}\left(t\right)$ is the compensated magnetic vector in the earth frame. Using this compensated magnetic vector, the gyroscope error can be corrected further.

After estimating the ${}_{E}^{S}{\hat{q}}_{est}\left(t\right)$, the Euler’s angle can be achieved by using the following equation,
(37)$${}_{E}^{S}{\hat{q}}_{est}\left(t\right)=\left[q_{1}^{{}^{\prime }}\left(t\right)\;\;q_{2}^{{}^{\prime }}\left(t\right)\;\;q_{3}^{{}^{\prime }}\left(t\right)\;\;q_{4}^{{}^{\prime }}\left(t\right)\right]$$
(38)$$\psi \left(t\right)=\tan^{-1}\left(\frac{2(q_{1}^{{}^{\prime }}\left(t\right)q_{4}^{{}^{\prime }}\left(t\right)+q_{2}^{{}^{\prime }}\left(t\right)q_{3}^{{}^{\prime }}\left(t\right))}{q_{1}^{{}^{\prime }2}\left(t\right)+q_{2}^{{}^{\prime }2}\left(t\right)-q_{3}^{{}^{\prime }2}\left(t\right)-q_{4}^{{}^{\prime }2}\left(t\right)}\right)$$
(39)$$\delta \left(t\right)=\sin^{-1}\left(2q_{1}^{{}^{\prime }}\left(t\right)q_{3}^{{}^{\prime }}\left(t\right)-2q_{2}^{{}^{\prime }}\left(t\right)q_{4}^{{}^{\prime }}\left(t\right)\right)$$
(40)$$\phi \left(t\right)=\tan^{-1}\left(\frac{2(q_{1}^{{}^{\prime }}\left(t\right)q_{2}^{{}^{\prime }}\left(t\right)+q_{3}^{{}^{\prime }}\left(t\right)q_{4}^{{}^{\prime }}\left(t\right))}{q_{1}^{{}^{\prime }2}\left(t\right)-q_{2}^{{}^{\prime }2}\left(t\right)-q_{3}^{{}^{\prime }2}\left(t\right)+q_{4}^{{}^{\prime }2}\left(t\right)}\right)$$
where $q_{1}^{{}^{\prime }}\left(t\right)$, $q_{2}^{{}^{\prime }}\left(t\right)$, $q_{3}^{{}^{\prime }}\left(t\right)$ and $q_{4}^{{}^{\prime }}\left(t\right)$ are components of ${}_{E}^{S}{\hat{q}}_{est}\left(t\right)$, ψ(t), δ(t) and ϕ(t) are yaw angle, pitch angle and roll angle of the NWMR, respectively. Because the NWMR moves on the topographic plane, the yaw angle is acted as the direction angle of the robot, therefore the roll angle and pitch angle are omitted.

### 4.2. Kalman Filter for the GPS Module

The Kalman filter is used to estimate the position of the NWMR and hand-held device from the output data of GPS. This filter is an optimal recursive data processing algorithm to minimize the mean squared error between the actual data and estimated data. The flow chart of Kalman filter is shown in Figure 9, the algorithm works in a two-step process. In prediction step, it predicts the current state variables. In update step, the difference between predicted value and measured value obtained from the sensor readout is taken into consideration. Kalman gain is counted for estimating the new predicted value and new uncertainty variance.

Prediction Step:
(41)$$\begin{array}{cc} \; & X_{p}\left(t\right)=FX_{e}\left(t-1\right)\\ \; & {\mathrm{Cov}}^{{}^{\prime }}\left(t\right)=F\mathrm{Cov}\left(t-1\right)F^{T}+Q_{\mathrm{distb}} \end{array}$$
where the current state variables matrix $X_{p}\left(t\right)$ is derived from its previous estimate value $X_{e}\left(t-1\right)$ and F is state transient matrix. ${\mathrm{Cov}}^{{}^{\prime }}\left(t\right)$ is prior error covariance matrix and $Q_{\mathrm{distb}}$ is the expect value of the system disturbance.

Kalman Gain:
(42)$$\begin{array}{cc} \; & K_{g}\left(t\right)=\frac{{\mathrm{Cov}}^{{}^{\prime }}\left(t\right)H^{T}}{{\mathrm{HCov}}^{{}^{\prime }}\left(t\right)H^{T}+R_{\mathrm{error}}} \end{array}$$
where $K_{g}\left(t\right)$ is the Kalman gain, H is the noiseless transition matrix from the real state variables and $R_{\mathrm{error}}$ is the expect value of the error measurement.

Update Step:
(43)$$\begin{array}{cc} \; & X_{e}\left(t\right)=X_{p}\left(t\right)+K_{g}\left(t\right)\left(Z_{\mathrm{in}}\left(t\right)-{\mathrm{HX}}_{p}\left(t\right)\right)\\ \; & \mathrm{Cov}\left(t\right)=\left(1-K_{g}\left(t\right)H\right){\mathrm{Cov}}^{{}^{\prime }}\left(t\right) \end{array}$$
where the update estimate $X_{e}\left(t\right)$ is derived from the measured data $Z_{\mathrm{in}}$. Cov(t) is the error covariance of the estimate datas.

The advantage of the Kalman filter is that the algorithm uses only the current measured data, the previously calculated state and its error covariance matrix, the past information is not needed. It prevents from the requirement of additional memory for storing the history data.

## 5. Simulations and Experimental Results

In this section, the simulations and the experiments are illustrated to demonstrate the merit and the effectiveness of the developed NWMR. Section 5.1 describes the hardware of the implemented NWMR and the hand-held device. In Section 5.2, the comparison results for the tracking performance of the controllers proposed by [22,23] are shown via MATLAB software. The experiments of trajectory tracking and dynamic real-time tracking of NWMR are demonstrated in Section 5.3.

### 5.1. Hardware Implementations

Figure 10a shows the physical control system of the NWMR. Frame marked with number 1 is the core control board of the NWMR; Frame 2 is the power switching board which is used to activate the NWMR and frame 3 is the relay board which is used as brakes to stop the NWMR. Frame 4 is motor driver board and it is used to drive the two wheels of NWMR. Figure 10b shows the lateral view of the core control board.

In the proposed NWMR, the part number MPU-9250 is used for 9-axis IMU, NEO-6M V2 is for GPS module and APC-220 is for RF transceiver. The STM32F429 embedded system is equipped under the core control board. Figure 11a,b shows the mechanism of the NWMR. Frame 1 of Figure 11a is the control system of the NWMR and frame 2 is the motors and battery modules. The hand-held device is shown in Figure 12. The blue bottom is utilized to activate the hand-held device.

### 5.2. Simulation Results

This section describes the simulation results of the tracking controller. For simplicity, the physical phenomenon presented in the real world, i.e., the friction of NWMR, the error range from the GPS modules which are mounted on the hand-held device and the NWMR, respectively, are ignored, therefore, the conceptual comparisons of the tracking capability for the tracking controllers in [22,23] are conducted. Three tracking cases are illustrated to compare the tracking performance of the controllers. First case is straight line tracking with linear velocity and second case is trajectory tracking of the circle with linear velocity and linear angular velocity. Third case is the random dynamic tracking example. As mentioned in Section 3.2, in order to achieve the controller to drive the trajectories asymptotically stable, the parameters of the controller are set as $k_{1}=2$, $k_{2}=4$, $k_{3}=10$ and $k_{4}=2$.

Figure 13 shows the tracking trajectory of the two controllers. The Reference straight line can be treated as the trajectory of the hand-held device. The initial posture of the hand-held device $P_{d}\left(0\right)=\left[x_{d},\;y_{d},\;\theta_{d}\right]$ is set to $\left[4,\;0,\;\frac{\pi }{4}\right]$, the initial linear velocity $v_{d}\left(0\right)$ is 4 m/s and the angular velocity $\omega_{d}$ is 0rad/s. The initial posture of the two tracking controllers $P_{c}\left(0\right)$ is set to $\left[-4,\;8,\;-\frac{\pi }{4}\right]$.

Figure 14a shows the posture error $x_{e}$ in X-axis. We set the convergent point at 0.03275, the figure shows that the convergence time of the controller proposed by [23] is 0.5293 sec, the convergence time of the controller by [22] is 1.891 sec. Figure 14b shows the posture error $y_{e}$ in Y-axis. By setting the convergent point as −0.00673, the convergence time of 3.177 sec and 3.595 sec can be obtained by using [22,23], respectively.

Figure 14c illustrates the angular error $\theta_{e}$. For the convergent point 0.004761, the figure shows that the convergence time of the controller in [23] is 1.824 s, the convergence time of the controller in [22] is 3.345 s.

Figure 15 shows the simulation result of the trajectory tracking of the circle. The initial posture of hand-held device $P_{d}\left(0\right)$ is set as $\left[4,\;0,\;\frac{\pi }{5}\right]$, the initial linear velocity $v_{d}\left(0\right)$ is 5m/s and the angular velocity $\omega_{d}$ is 1 rad/s. The initial posture of two tracking controllers $P_{c}\left(0\right)$ is set to [−4,−1,0]. Figure 16a shows the posture error $x_{e}$ in X-axis. As seen in the figure, for the convergent point −0.01604, the convergence time of the controller [23] is 2.692 s, the convergence time of the controller [22] is 3.01 s. Figure 16b shows the posture error $y_{e}$ in Y-axis. We set the convergent point at 0.003297, the figure shows that the convergence time of the controller proposed by [23] is 2.952 s, the convergence time of the controller proposed by [22] is 3.811 s. Figure 16c illustrates the angular error $\theta_{e}$. For the convergent point of −0.0004928, the figure shows that the convergence time of the controller by [23] is 3.969 s, the convergence time of the controller by [22] is 4.328 s.

From the simulation results of straight line tracking and trajectory tracking of the circle, it can readily seen that the controller proposed by [23] can provide a better response than the controller proposed by [22].

In order to show the dynamic real-time tracking, the $\theta_{d}$ of the tracking object in (Equation 5) needs be continuously updated. For the Consideration of the four-quadrant angle conversion, the actual angle $\theta_{d\_update}$ is recalculated by the following equation,
(44)$$\begin{array}{cc} \; & \theta_{d\_update}=90^{\circ }-\theta_{d}\;\;\;if\;\;0^{\circ }<\theta_{d}<90^{\circ }\\ \; & \theta_{d\_update}=90^{\circ }+\theta_{d}\;\;\;if\;\;90^{\circ }<\theta_{d}<180^{\circ }\\ \; & \theta_{d\_update}=270^{\circ }+\theta_{d}\;\;\;if\;\;180^{\circ }<\theta_{d}<270^{\circ }\\ \; & \theta_{d\_update}=270^{\circ }-\theta_{d}\;\;\;if\;\;270^{\circ }<\theta_{d}<360^{\circ } \end{array}$$

The linear velocity of the tracking object $u_{d}$ in (15) also needs be continuously updated and the actual linear velocity $v_{d\_update}$ is
(45)$$v_{d\_update}=\frac{\sqrt{{\left(x_{d}\left(t\right)-x_{d}\left(t-1\right)\right)}^{2}+{\left(y_{d}\left(t\right)-y_{d}\left(t-1\right)\right)}^{2}}}{\Delta t}$$
where $x_{d}\left(t\right)$ and $x_{d}\left(t-1\right)$ are the positions toward X direction at time *t* and t−1, respectively. $y_{d}\left(t\right)$ and $y_{d}\left(t-1\right)$ are the positions toward Y direction at time *t* and t−1, respectively, Δt is sampled time. Because the tracking object is treated as a point in the world X-Y coordinate, the angular velocity of the hand-held device $\omega_{d}$ in (15) is set to zero for simplicity.

Figure 17a shows the simulation result of the dynamic real-time tracking of the tracking controller proposed by [23]. The initial position of hand-held device $\left[x_{d},\;y_{d}\right]$ is [4,1], the initial linear velocity $v_{d}\left(0\right)$ is 3m/s and the angular velocity $\omega_{d}$ is 0rad/s. The initial posture of NWMR $P_{c}\left(0\right)$ is [2,0,0]. Figure 17b shows the posture error of the NWMR with respect to the hand-held device, the angular error is converged to 0.003546 after 3.928 s.

In order to implement the NWMR, the tracking controller and algorithms mentioned above are all implemented with C language in MDK-ARM integrated development environment.

### 5.3. Experiments

The processes of the trajectory tracking of the NWMR are shown in Figure 18a–i. The video is attached in Appendix A. We first set the initial postures of NWMR and the hand-held device, the user holds the hand-held device and is acted as tracking object. Figure 18a–i show the trajectory tracking responses of the developed NWMR. The robot moves toward to the user and stop while the distance between the NWMR and objective is smaller than 1.5 m.

The dynamic real-time tracking of the NWMR are shown in Figure 19a–l. The video is attached in Appendix A. These figures show the processes of real-time tracking. From Figure 19a–l, it can be seen that the real-time tracking can be achieved by the developed NWMR. When the user goes forward and go backward, the NWMR can track the hand-held device smoothly. Even if the user turns right and turns left, the robot still can track accurately.

## 6. Conclusions and Future Works

In this paper, the trajectory tracking for the NWMR is investigated. GPS and 9-axis IMU sensor are utilized to measure the posture and position of the NWMR and target device. For the consideration of the small posture error and fast convergence in the trajectory tracking, Kalman filter and Madgwick’s fusion algorithm are used. The tracking controller with these algorithms are implemented by an embedded system for fast prototype and cost efficiency. The simulations and the experiment results show the feasibility and the effectiveness of the proposed NWMR.

For designing the next generation of the NWMR, obstacle avoidance, pattern recognition, multi-sensor fusion algorithms and power management strategy are four main research topics. For the outdoor application, a common GPS will be replaced by the RTK-GPS due to error range from 20 cm to 2 m. For the obstacle avoidance approach, millimeter-Wave (mmWave) radar sensor and Light-detection and ranging (LIDAR) sensor will be used. The advantage of the mmWave radar sensor is impervious to environmental conditions such as rain, fog and dust. The LIDAR sensor has the advantage of high resolution for a short distance measurement. Pattern recognition module integrated image sensor, optical lens and image processor is used to identify objects by utilizing the machine learning algorithms. Moreover, in order to extend the operating time of NWMR, power management strategy and will be studied further.

## Figures and Tables

**Figure 1 sensors-20-07055-f001:**
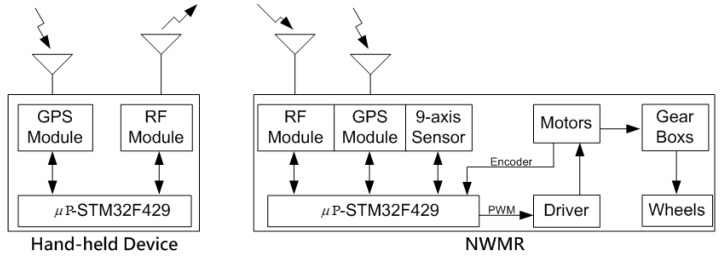
The block diagram of the proposed NWMR system.

**Figure 2 sensors-20-07055-f002:**
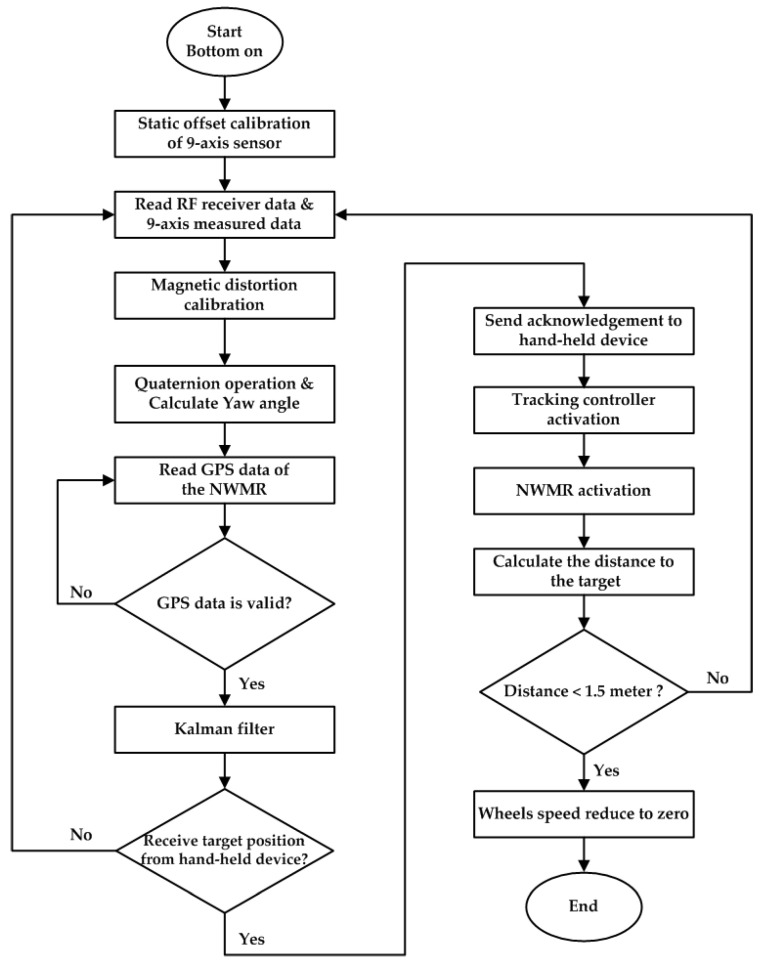
The flow chart of the trajectory tracking for the NWMR.

**Figure 3 sensors-20-07055-f003:**
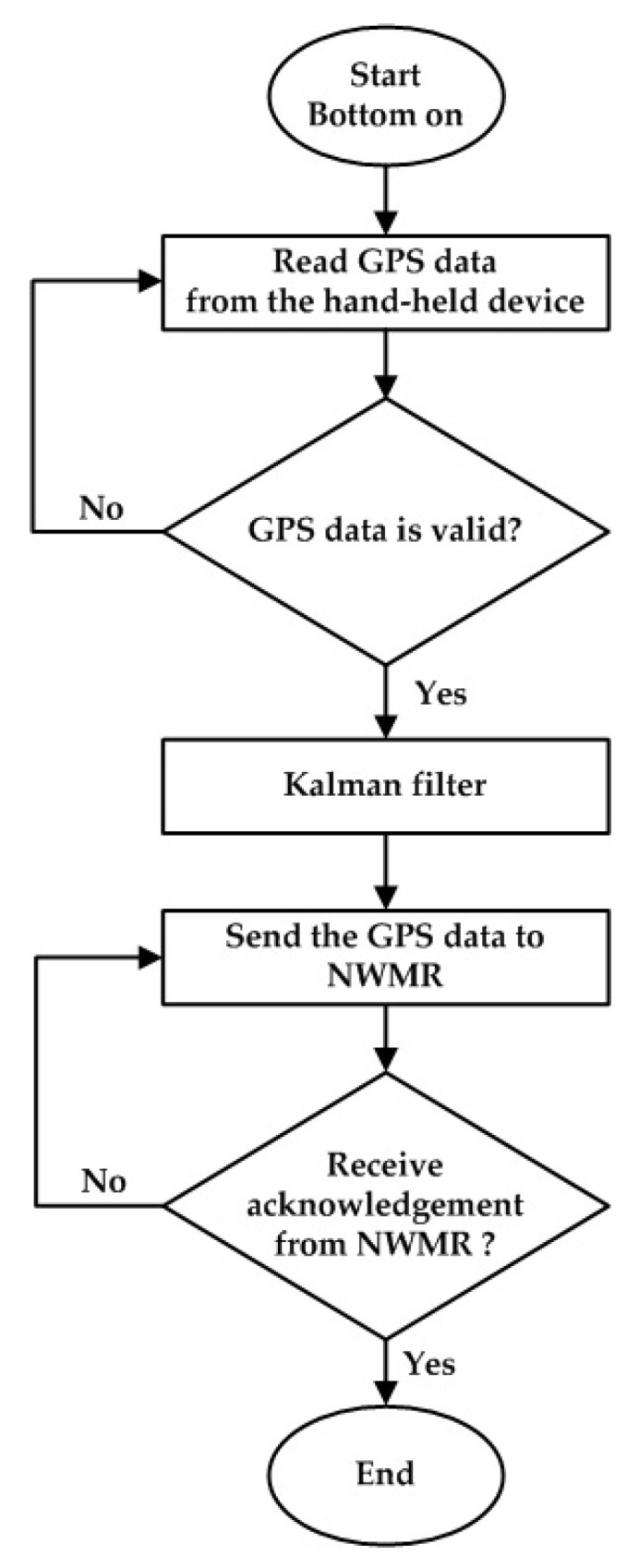
The flow chart of the trajectory tracking for the hand-held device.

**Figure 4 sensors-20-07055-f004:**
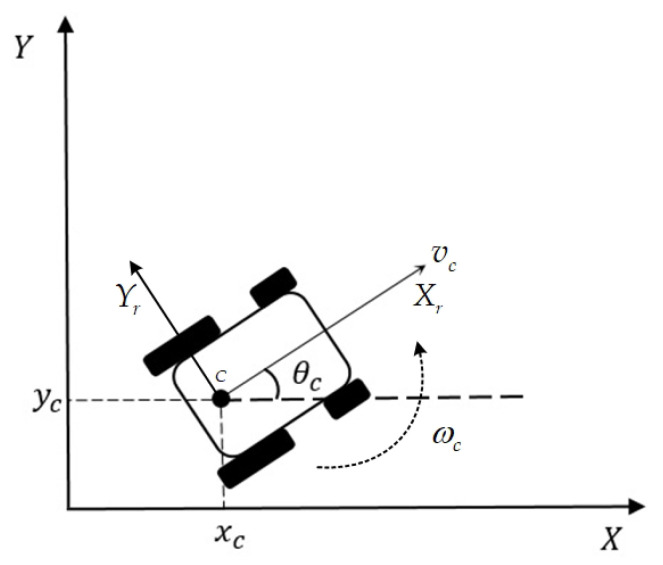
The posture of the NWMR.

**Figure 5 sensors-20-07055-f005:**
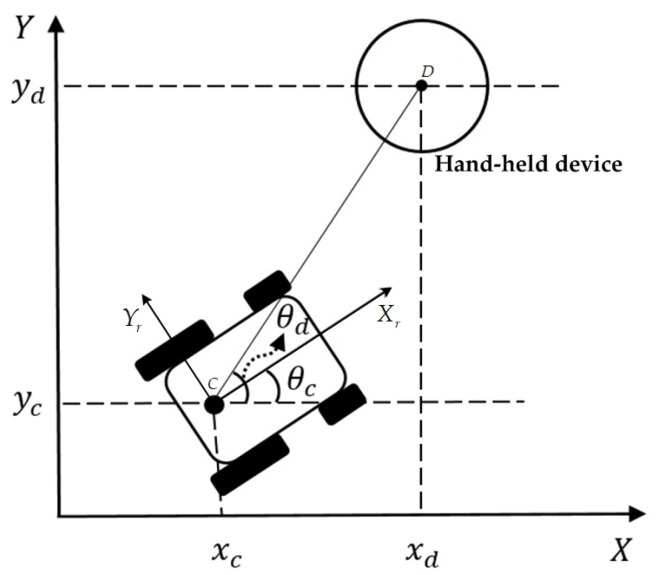
The posture of the hand-held device.

**Figure 6 sensors-20-07055-f006:**
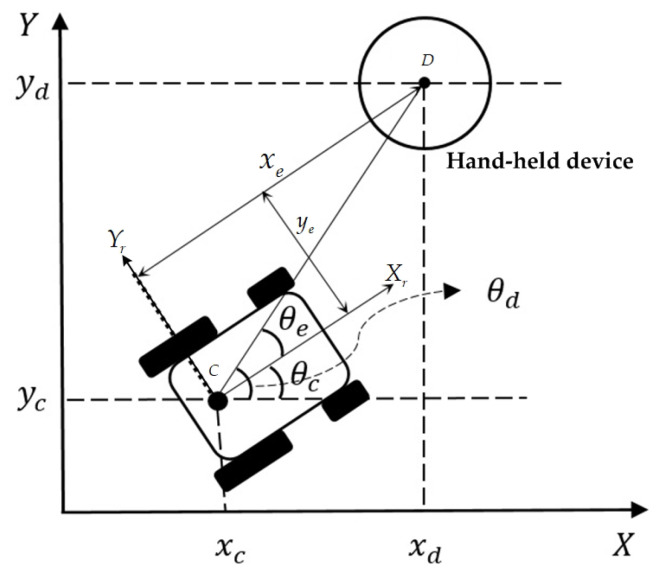
The posture error of the NWMR and the hand-held device.

**Figure 7 sensors-20-07055-f007:**
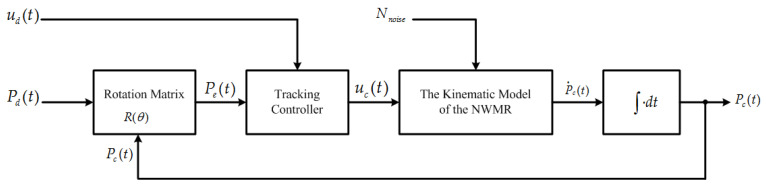
The function block of the tracking controller design.

**Figure 8 sensors-20-07055-f008:**
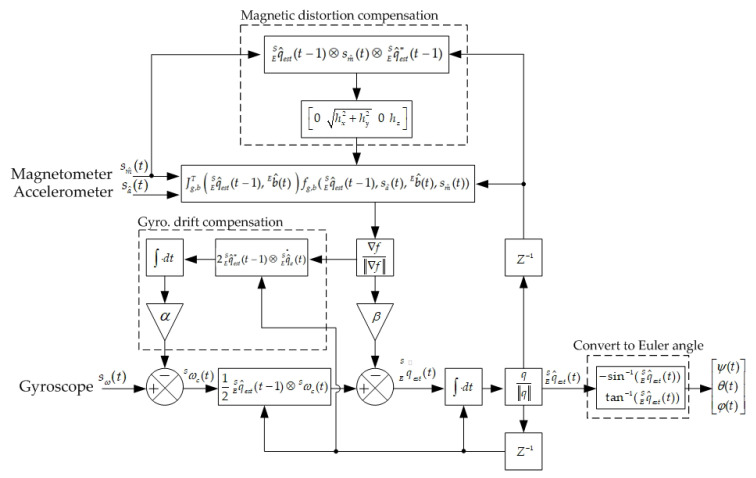
The block diagram of Madgwick’s data fusion algorithm for the 9-axis IMU Sensor.

**Figure 9 sensors-20-07055-f009:**
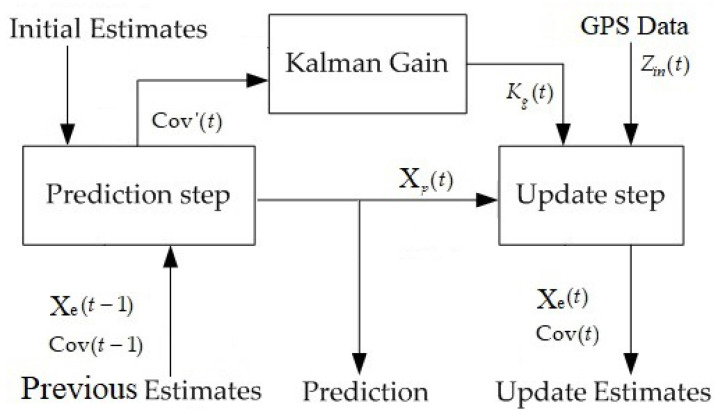
Kalman filter recursive algorithm.

**Figure 10 sensors-20-07055-f010:**
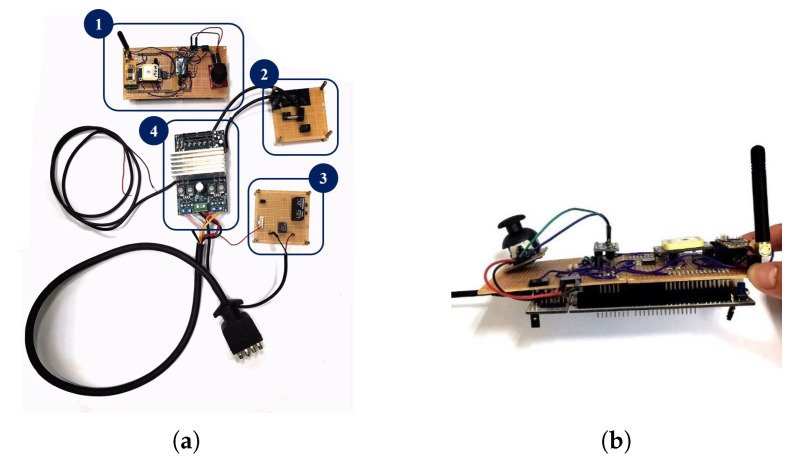
The overall view of the control system of the NWMR: (**a**) The physical control system of the NWMR; (**b**) The lateral view of the core control board of the NWMR.

**Figure 11 sensors-20-07055-f011:**
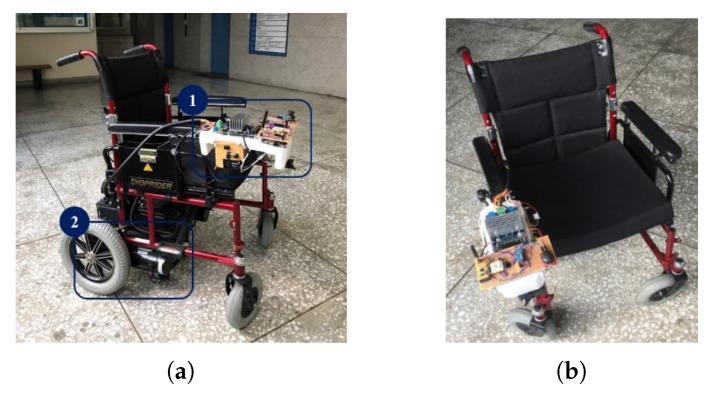
The overall view of the NWMR: (**a**) The lateral view; (**b**) The top view.

**Figure 12 sensors-20-07055-f012:**
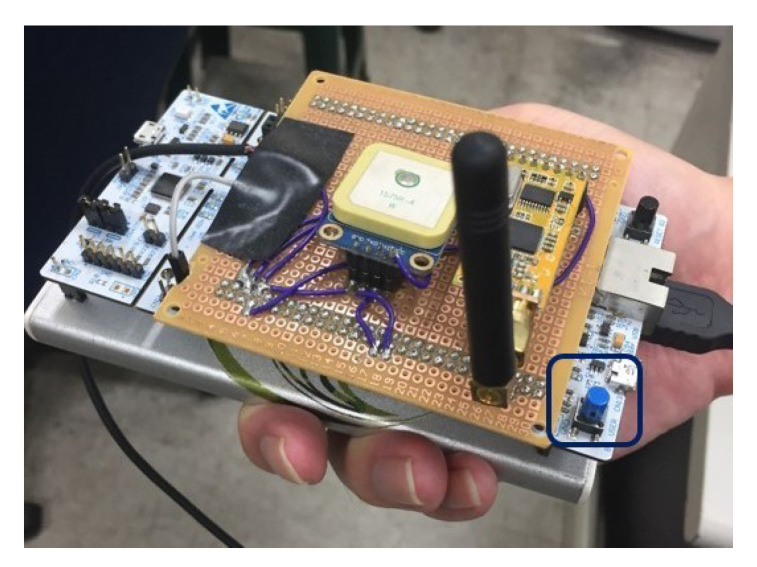
The physical view of the hand-held device.

**Figure 13 sensors-20-07055-f013:**
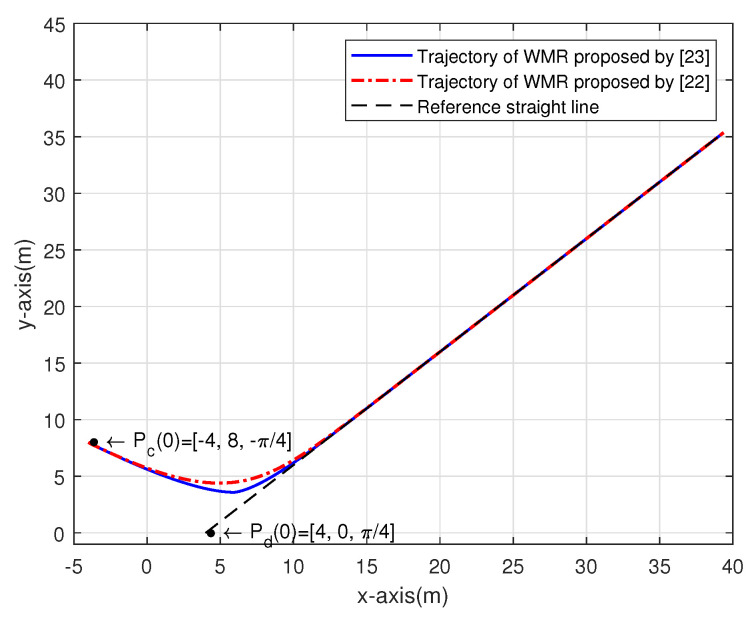
The trajectory of the straight line tracking.

**Figure 14 sensors-20-07055-f014:**
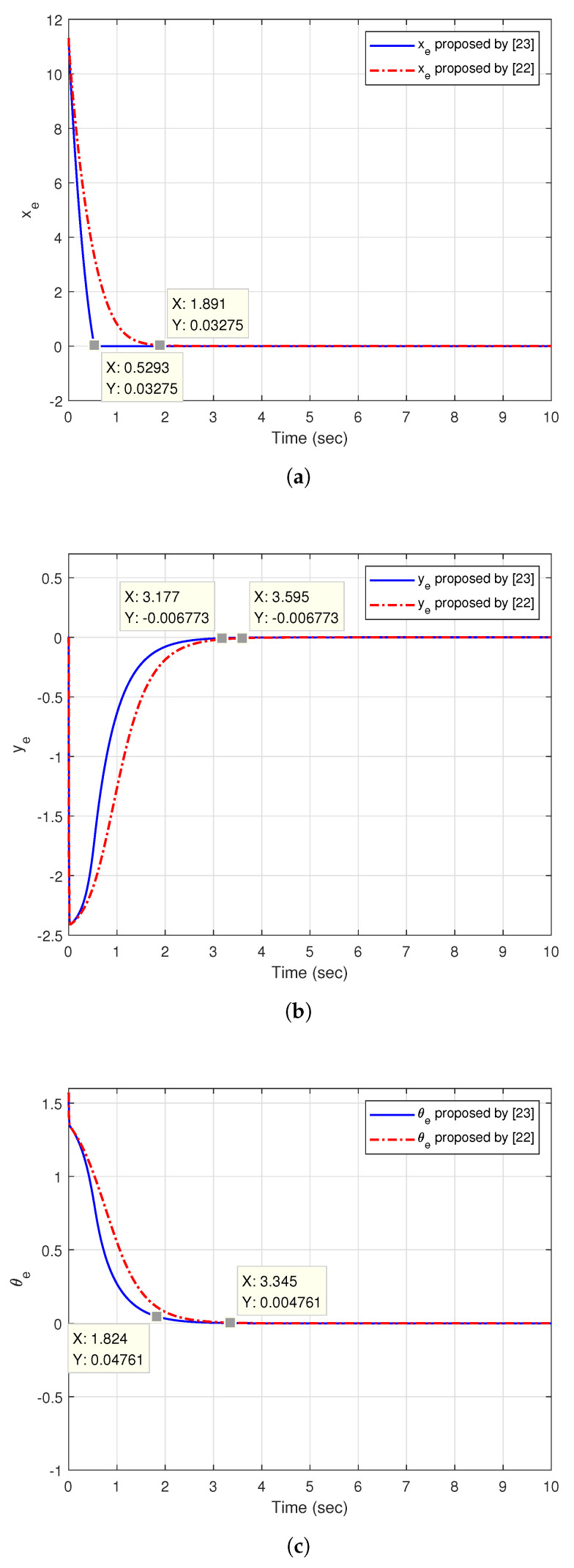
The posture errors of straight line tracking.: (**a**) $x_{e}$; (**b**) $y_{e}$; (**c**) $\theta_{e}$.

**Figure 15 sensors-20-07055-f015:**
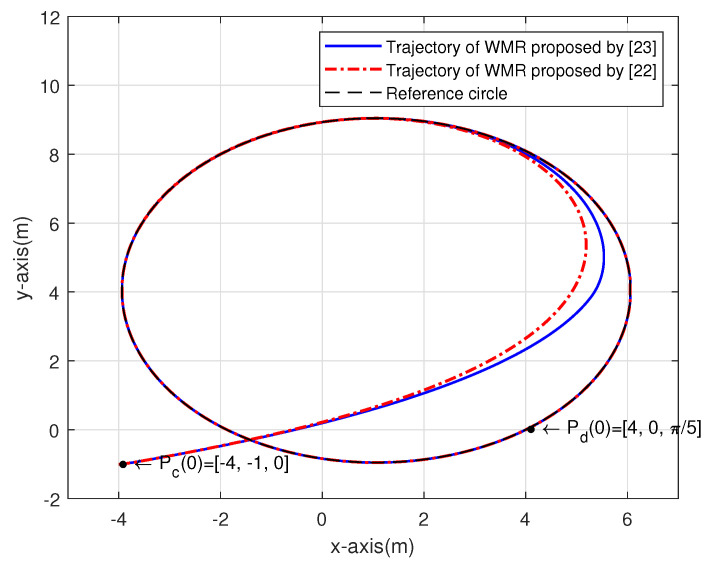
The trajectory tracking of the circle.

**Figure 16 sensors-20-07055-f016:**
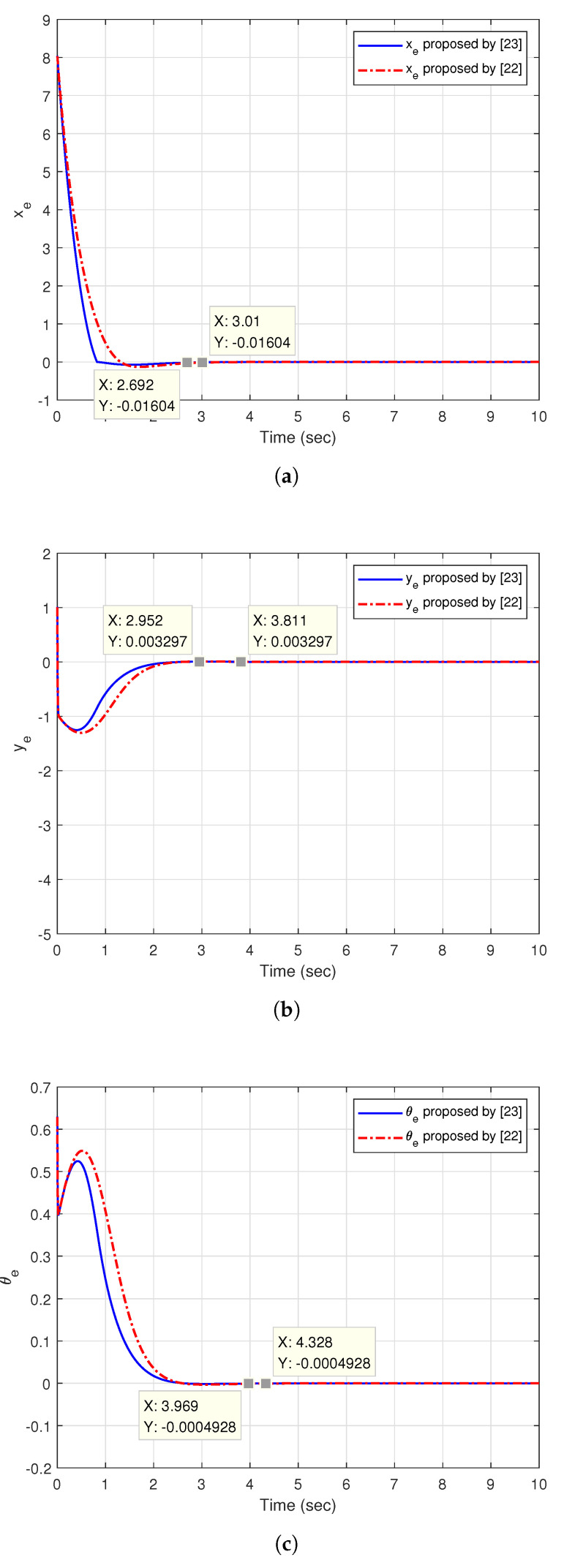
The posture error of trajectory tracking of the circle: (**a**) $x_{e}$; (**b**) $y_{e}$; (**c**) $\theta_{e}$.

**Figure 17 sensors-20-07055-f017:**
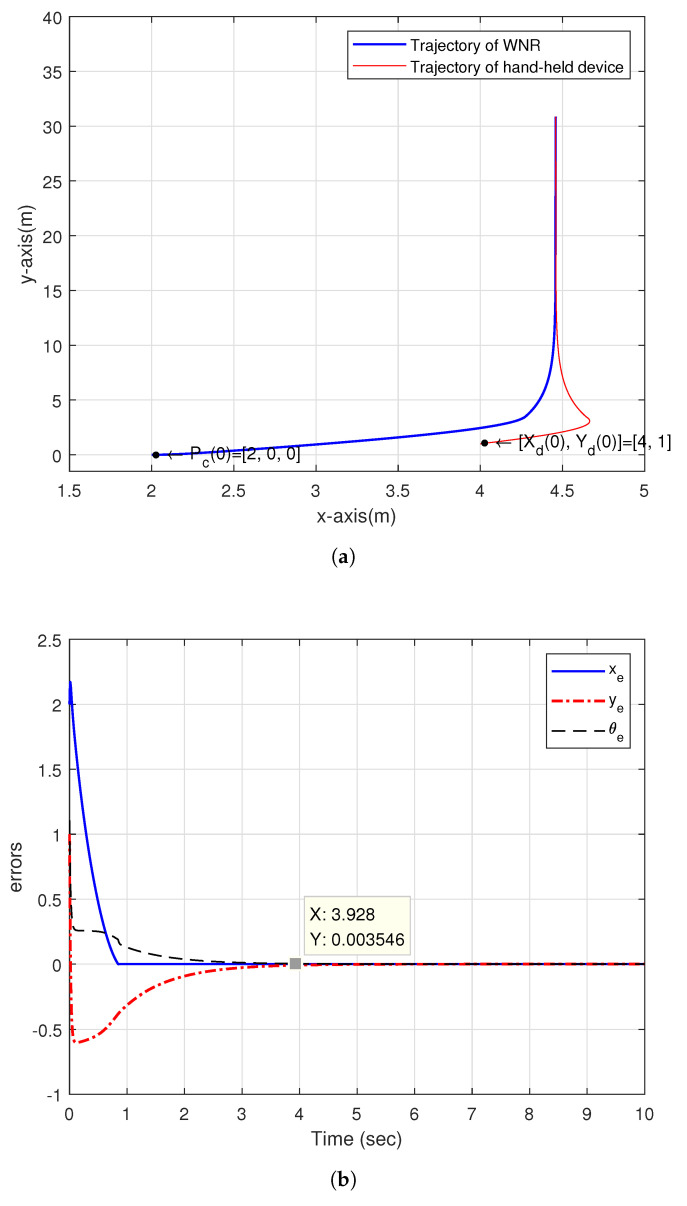
The dynamic real-time tracking: (**a**) Trajectory between NWMR and hand-held device; (**b**) The posture error for the dynamic real-time tracking.

**Figure 18 sensors-20-07055-f018:**
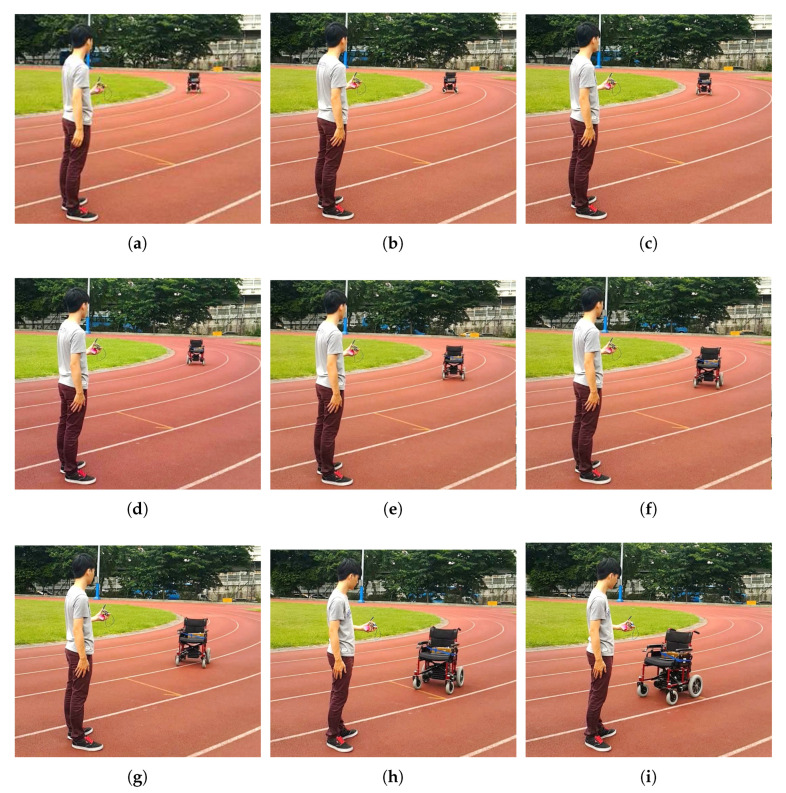
The processes of the trajectory tracking.

**Figure 19 sensors-20-07055-f019:**
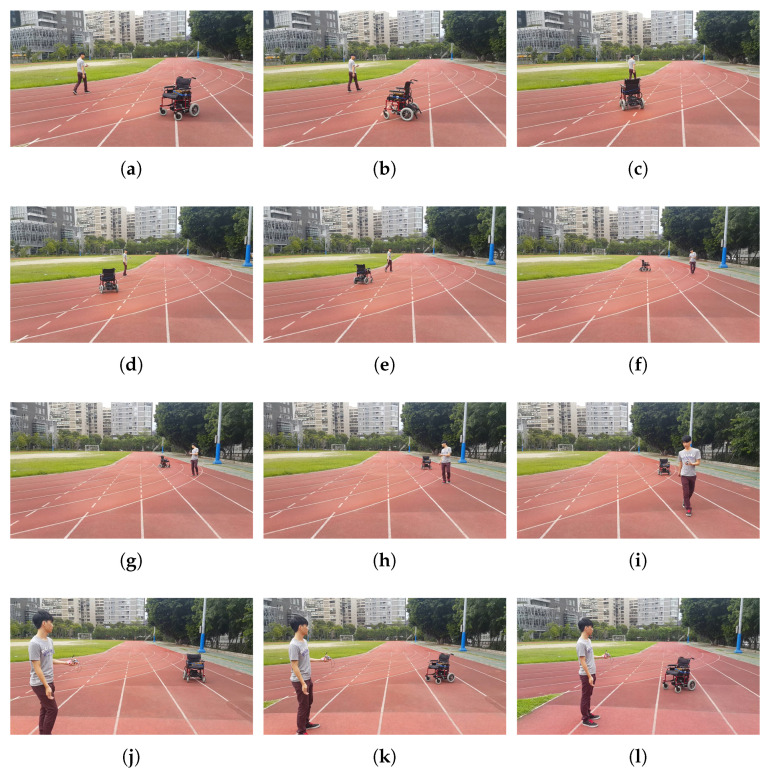
The processes of dynamic real-time tracking.

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
