# Peer review of "A Sensor Fusion Based Nonholonomic Wheeled Mobile Robot for Tracking Control"

_sensors, 2020, doi:10.3390/s20247055_

Round 1

Reviewer 1 Report

1.- In Section 3. Please state the stability properties: local, global, asymptotic, conditions to be satisfied by k1,k2,k3,k4 and vd.

2.- Section 4.1. The result with quaternions is a contribution of the paper? or is an application of  some previous result result in the literature? Please state the conditions that must be satisfied to  allow the result to work.

3.- Kalman filter methodology is a well known strategy. Please, avoid the presentation of a long procedure, just present the equations to be computed on line and conditions that must be satified to ensure the estimation process to work.

4.- What is the main contribution of the paper? I mean, several practical works already exist on this topic. What is new in your proposal?

Reviewer 2 Report

The paper describes a tracking algorithm for a wheel chair that is able to follow a hand-held GPS receiver. The paper is very well written and includes a very nice and good analysis of the state of the art.

The proposed solution is based on a conventional Kalman Filter which is itself based on a previous publication. The approach and all relevant information is described in detail and can be understood very well.

In the experimental section the hardware is described but the type of the GPS and inertial sensors are not mentioned. Several experiments with line, circle and dynamic tracking are presented based on a simulation. For the reader it is unclear why a simulation environment is suitable for the evaluation of the approach.

The final experiments with the physical wheel chair are very interesting and nice to see (including the additional videos) but it is left unclear how these examples can prove the efficiency of the approach. A comparison with any already existing solution is missing.

Although the paper is very interesting and informative it is unclear what the scientific goal of the paper is. A qualitative comparison of the results is missing and therefore it is not possible to judge the quality of the presented method. 

Reviewer 3 Report

The manuscript proposes the development of a nonholonomic wheeled mobile robot for real-time trajectory tracking based on IMU and GPS. The authors used a Kalman filter to filter the noise of the sensors. The work is interesting. However, the references are too old. Please, update the references to more recent ones (2019, 2020, and 2021). Besides, there are no articles from the Sensors database to show that the paper fits this journal. My main concern is the fact that I did not identify any innovations. This could be because the authors did not explore related works and their tracking controller in order to compare with the proposing one.

1 - The manuscript needs an in-depth English review. Please, provide it.
2- The authors should highlight their main contributions. Please, use bullets for this. Also, in the introduction, provide more background to the problem context. Why is your work scientifically relevant?
3 - It is not possible to say if this work is in the state-of-the-art if there is no comparison with other related works. Please, provide more related works in the Introduction Section.
4 - The authors cited a few papers working with NWMR and different control strategies. However, what these mentioned works are related to the proposed work? Please, cite papers that use similar control strategies and similar sensors (there is plenty of works).
5 - Line 40, where is the reference for Dong?
As far as I know, Madgwick sensor fusion is a library already done for some microcontrollers. Is this already implemented in the STN32F429? I mean that the authors did not implement it. The block diagram of Figure 8 is the Madgwick that is already implemented in the microcontroller? Or the authors modify anything? This should be explicit in this manuscript.
6 – The authors used a simple Kalman filter to reduce error from GPS. However, a common GPS has errors that can range from 20 cm to 2 meters or more. This simple filter will not solve this problem. I recommend using other strategies, such as Extended Kalman Filter and other sensors, or an RTK-GPS.
9 – Put the background of Figure 10 (a) white.
10 – The authors only presented qualitative results. Please, provide more quantitative results.
11 – What is the processing time of your algorithm?
11- The authors must provide comparisons with other works in the discussions of their results.
12 – The section Conclusions should be renamed to “Conclusions and Future Work,” and the authors should show future works ideas.

Round 2

Reviewer 2 Report

The authors have very carefully reacted to the mentioned topics and corrected them accordingly. The improvements of the paper is considered good.

The comparison of the tracking performance of the simulation is much better described and the graphical representation is significantly improved.

Still there are two more comments which require a minor revision

  1. Add a sentence why a comparison of tracking algorithms in simulation is also applicable to real world scenarios ? Is the simulation sufficiently close to real situations or is a conceptual comparison ?
  2. (I think I have given this comment before) Website links are not considered as references and must be removed from the list of references (either footnotes or in text). These are 31, 33 and 34. This is mandatory from my point of view.  

Reviewer 3 Report

The manuscript has significantly improved. Regarding the English language, I recommend a minor spell check.
